# Differentially Private Bayesian Linear Regression

**Garrett Bernstein**
University of Massachusetts Amherst
gbernstein@cs.umass.edu

**Daniel Sheldon**
University of Massachusetts Amherst
sheldon@cs.umass.edu

## Abstract

Linear regression is an important tool across many fields that work with sensitive human-sourced data. Significant prior work has focused on producing differentially private point estimates, which provide a privacy guarantee to individuals while still allowing modelers to draw insights from data by estimating regression coefficients. We investigate the problem of Bayesian linear regression, with the goal of computing posterior distributions that correctly quantify uncertainty given privately released statistics. We show that a naive approach that ignores the noise injected by the privacy mechanism does a poor job in realistic data settings. We then develop noise-aware methods that perform inference over the privacy mechanism and produce correct posteriors across a wide range of scenarios.

## 1 Introduction

Linear regression is one of the most widely used statistical methods, especially in the social sciences [Agresti and Finlay, 2009] and other domains where data comes from humans. It is important to develop robust tools that can realize the benefits of regression analyses but maintain the privacy of individuals. *Differential privacy* [Dwork et al., 2006] is a widely accepted formalism to provide algorithmic privacy guarantees: a differentially private algorithm randomizes its computation to provably limit the risk that its output discloses information about individuals.

Existing work on differentially private linear regression focuses on frequentist approaches. A variety of privacy mechanisms have been applied to point estimation of regression coefficients, including sufficient statistic perturbation (SSP) [Foulds et al., 2016, McSherry and Mironov, 2009, Vu and Slavkovic, 2009, Wang, 2018, Zhang et al., 2016], posterior sampling (OPS) [Dimitrakakis et al., 2014, Geumlek et al., 2017, Minami et al., 2016, Wang, 2018, Wang et al., 2015, Zhang et al., 2016], subsample and aggregate [Dwork and Smith, 2010, Smith, 2008], objective perturbation [Kifer et al., 2012], and noisy stochastic gradient descent [Bassily et al., 2014]. Only a few recent works address uncertainty quantification through confidence interval estimation [Sheffet, 2017] and hypothesis tests [Barrientos et al., 2019] for regression coefficients.

We develop a differentially private method for *Bayesian* linear regression. A Bayesian approach naturally quantifies parameter uncertainty through a full posterior distribution and provides other Bayesian capabilities such as the ability to incorporate prior knowledge and compute posterior predictive distributions. Existing approaches to private Bayesian inference include OPS (see above), MCMC [Wang et al., 2015], variational inference (VI; Honkela et al. [2018], Jälkö et al. [2017], Park et al. [2016]), and SSP [Bernstein and Sheldon, 2018, Foulds et al., 2016], but none provide a fully satisfactory approach for Bayesian regression modeling. OPS does not naturally produce a representation of a full posterior distribution. MCMC approaches incur per-iteration privacy costs and satisfy only approximate $(\epsilon, \delta)$-differential privacy. Private VI approaches also incur per-iteration privacy costs, and are most relevant when the original inference problem requires VI. When applicable, SSP is a very desirable approach — sufficient statistics are perturbed once and then used in conjugate updates to obtain parameters of full posterior distributions — and often outperforms other methods in practice [Foulds et al., 2016, Wang, 2018]. However, Bernstein and Sheldon [2018] demonstrated

(for unconditional exponential family models) that naive SSP, which ignores noise introduced by the privacy mechanism, systematically underestimates uncertainty at small to moderate sample sizes. We show that the same phenomenon holds for Bayesian linear regression: naive SSP produces private posteriors that are properly calibrated asymptotically in the sample size, but for realistic data sets and privacy levels may need very large population sizes to reach the asymptotic regime.

This motivates our development of Bayesian inference methods for linear regression that properly account for the noise due to the privacy mechanism [Bernstein and Sheldon, 2018, Bernstein et al., 2017, Karwa et al., 2014, 2016, Schein et al., 2018, Williams and McSherry, 2010]. We leverage a model in which the data and model parameters are latent variables, and noisy sufficient statistics are observed, and then develop MCMC-based techniques to sample from posterior distributions, as done for exponential families in [Bernstein and Sheldon, 2018]. A significant challenge relative to prior work is the handling of covariate data. Typical regression modeling treats only response variables and parameters as random, and conditions on covariates. This is not possible in the private setting, where covariates must be kept private and therefore treated as latent variables. We therefore require some form of assumption about the distribution over covariates. We develop two inference methods. The first includes latent variables for each individual; it requires an explicit prior distribution for covariates and its runtime scales with population size. The second marginalizes out individuals and approximates the distribution over the sufficient statistics; it requires weaker assumptions about the covariate distribution (only moments), and its running time does not scale with population size. We perform a range of experiments to measure the calibration and utility of these methods. Our noise-aware methods are as well or nearly as well calibrated as the non-private method, and have better utility than the naive method. We demonstrate using real data that our noise-aware methods quantify posterior predictive uncertainty significantly better than naive SSP.

## 2 Background

**Differential Privacy.** A differentially private algorithm $\mathcal{A}$ provides a guarantee to individuals: The distribution over the output of $\mathcal{A}$ will be (nearly) indistinguishable regardless of the inclusion or exclusion of a single individual's data. The implication to the individual is they face negligible risk in deciding to contribute their personal data to be used by a differentially private algorithm. To formally write the guarantee we reason about a generic data set $X = x_{1:n} = (x_1, \cdots, x_n)$ of $n$ individuals, where $x_i$ is the data record of the $i$th individual. For this paper, define *neighboring* data sets as those that differ by a single record, i.e. $X' \in \text{nbrs}(X)$ if $X' = (x_{1:i-1}, x_i', x_{i+1:n})$ for some $i$. [1]

**Definition 1** (Differential Privacy; Dwork et al. [2006]). *A randomized algorithm $\mathcal{A}$ satisfies $\epsilon$-differential privacy if for any input $X$, any $X' \in \text{nbrs}(X)$ and any subset of outputs $O \subseteq \text{Range}(\mathcal{A})$, $\Pr[\mathcal{A}(X) \in O] \leq \exp(\epsilon)\Pr[\mathcal{A}(X') \in O].$*

The above guarantee is ensured by randomizing $\mathcal{A}$. A key concept is the *sensitivity* of a function, which quantifies the impact an individual record has on the output of the function.

**Definition 2** (Sensitivity; Dwork et al. [2006]). *The* sensitivity *of a function $f$ is $\Delta_f = \sup_{X,X' \in \text{nbrs}(X)} \|f(X) - f(X')\|_1$.*

We use the *Laplace mechanism* to ensure publicly-released statistics meet the requirements of differential privacy.

**Definition 3** (Laplace Mechanism; Dwork et al. [2006]). *Given a function $f$ that maps data sets to $\mathbb{R}^m$, the Laplace mechanism outputs the random variable $\mathcal{L}(X) \sim \text{Lap}\left(f(X), \Delta_f/\epsilon\right)$ from the Laplace distribution, which has density $\text{Lap}(z; u, b) = (2b)^{-m} \exp\left(-\|z - u\|_1/b\right)$. This corresponds to adding zero-mean independent noise $u_i \sim \text{Lap}(0, \Delta_f/\epsilon)$ to each component of $f(X)$.*

A final property is *post-processing*, which says that any further processing on the output of a differentially private algorithm that does not access the original data retains the same privacy guarantees [Dwork and Roth, 2014].

**Linear Regression.** We start with a standard (non-private) linear regression problem. An individual's *covariate* or *regressor* data is $\mathbf{x} \in \mathbb{R}^d$ and the dependent *response* data is $y \in \mathbb{R}$. We will assume a

conditionally Gaussian model $y \sim \mathcal{N}(\boldsymbol{\theta}^T \mathbf{x}, \sigma^2)$, where $\boldsymbol{\theta} \in \mathbb{R}^d$ are the regression coefficients and $\sigma^2$ is the error variance. An intercept or bias term may be included in the model by appending a unit-valued feature to $\mathbf{x}$. The goal, given an observed population of $n$ individuals, is to obtain a point estimate of $\boldsymbol{\theta}$. The population data can be written as $X \in \mathbb{R}^{n \times d}$, where each row corresponds to an individual $\mathbf{x}$, and $\mathbf{y} \in \mathbb{R}^n$. The ordinary least squares (OLS) solution is $\hat{\boldsymbol{\theta}} = \left(X^T X\right)^{-1} X^T \mathbf{y}$ [Rencher, 2003].

In Bayesian linear regression the parameters $\boldsymbol{\theta}$ and $\sigma^2$ are random variables with a specified prior distribution. The conjugate priors are $p(\sigma^2) = \text{InverseGamma}(a_0, b_0)$ and $p(\boldsymbol{\theta} \mid \sigma^2) = \mathcal{N}(\boldsymbol{\mu}_0, \sigma^2 \boldsymbol{\Lambda}_0^{-1})$, which defines a normal-inverse gamma prior distribution: $p(\boldsymbol{\theta}, \sigma^2) = \text{NIG}(\boldsymbol{\mu}_0, \boldsymbol{\Lambda}_0, a_0, b_0)$. Due to conjugacy of the prior distribution with the likelihood model, the posterior distribution, shown in Equation (1), is also normal-inverse gamma [O'Hagan and Forster, 1994].

$$
\begin{aligned}
p(\boldsymbol{\theta}, \sigma^2 \mid X, \mathbf{y}) &= \text{NIG}(\boldsymbol{\mu}_n, \boldsymbol{\Lambda}_n, a_n, b_n) \\
\boldsymbol{\mu}_n &= \left(X^T X + \boldsymbol{\Lambda}_0\right)^{-1} \left(X^T \mathbf{y} + \boldsymbol{\mu}_0^T \boldsymbol{\Lambda}_0\right) \\
\boldsymbol{\Lambda}_n &= X^T X + \boldsymbol{\Lambda}_0 \\
a_n &= a_0 + \frac{1}{2} n \\
b_n &= b_0 + \frac{1}{2} \left(\mathbf{y}^T \mathbf{y} + \boldsymbol{\mu}_0^T \boldsymbol{\Lambda}_0 \boldsymbol{\mu}_0 - \boldsymbol{\mu}_n^T \boldsymbol{\Lambda}_n \boldsymbol{\mu}_n\right)
\end{aligned}
\tag{1}
$$

Let $t(\mathbf{x}, y) := [\text{vec}(\mathbf{x}\mathbf{x}^T), \mathbf{x}y, y^2]$ for an arbitrary individual. Then the sufficient statistics of the above model are $\mathbf{s} := t(X, \mathbf{y}) = \sum_i t(\mathbf{x}^{(i)}, y^{(i)}) = \left[X^T X, X^T \mathbf{y}, \mathbf{y}^T \mathbf{y}\right]$. These capture all information about the model parameters contained in the sample and are the only quantities needed for the conjugate posterior updates above [Casella and Berger, 2002].

# 3 Private Bayesian Linear Regression

The goal is to perform Bayesian linear regression in an $\epsilon$-differentially private manner. We ensure privacy by employing sufficient statistic perturbation (SSP) [Foulds et al., 2016, Vu and Slavkovic, 2009, Zhang et al., 2016], in which the Laplace mechanism is used to inject noise into the sufficient statistics of the model, making them fit for public release. The question is then how to compute the posterior over the model parameters $\boldsymbol{\theta}$ and $\sigma^2$ given the noisy sufficient statistics. We first consider a *naive* method that ignores the noise in the noisy sufficient statistics. We then consider more principled *noise-aware* inference approaches that account for the noise due to the privacy mechanism.

## 3.1 Privacy mechanism

Using the Laplace mechanism to release the noisy sufficient statistics $\mathbf{z}$ results in the model shown in Figure 1. This is the same model used in non-private linear regression except for the introduction of $\mathbf{z}$, which requires the exact sufficient statistics $\mathbf{s}$ to have finite sensitivity. A standard assumption in literature [Awan and Slavkovic, 2018, Sheffet, 2017, Wang, 2018, Zhang et al., 2012] is to assume $\mathbf{x}$ and $y$ have known a priori lower and upper bounds, $(a_\mathbf{x}, b_\mathbf{x})$ and $(a_y, b_y)$, with bound widths $w_\mathbf{x} = b_\mathbf{x} - a_\mathbf{x}$ (assuming, for simplicity, equal bounds for all covariate dimensions) and $w_y = b_y - a_y$, respectively. We can then reason about the worst case influence of an individual on each component of $\mathbf{s} = \left[X^T X, X^T \mathbf{y}, \mathbf{y}^T \mathbf{y}\right]$, recalling that $\mathbf{s} = \sum_i t(\mathbf{x}^{(i)}, y^{(i)})$, so that $\left[\Delta_{(X^T X)_{jk}}, \Delta_{(Xy)_j}, \Delta_{y^2}\right] = \left[w_\mathbf{x}^2, w_\mathbf{x} w_y, w_y^2\right]$. The number of unique elements[2] in $\mathbf{s}$ is $[d(d+1)/2, d, 1]$, so $\Delta_\mathbf{s} = w_\mathbf{x}^2 d(d+1)/2 + w_\mathbf{x} w_y d + w_y^2$. The noisy sufficient statistics fit for public release are $\mathbf{z} = \left[z_i \sim \text{Lap}(s_i, \Delta_\mathbf{s}/\epsilon) : s_i \in \mathbf{s}\right]$.

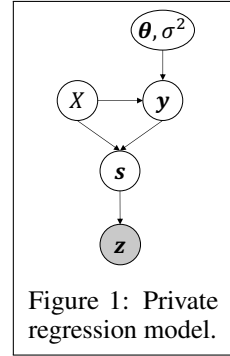

Figure 1: Private regression model.

## 3.2 Noise-naive method

Previous work developed methods to obtain OLS solutions via SSP by ignoring the noise injected into the sufficient statistics [Awan and Slavkovic, 2018, Sheffet, 2017, Wang, 2018]. One corresponding approach for Bayesian regression is to naively replace $\mathbf{s}$ in Figure 1 with the noisy version $\mathbf{z}$ and then perform the conjugate update in Equation (1). This noise-naive method (Naive) is simple and fast, and we empirically show in Section 4 that it produces an asymptotically correct posterior.

## 3.3 Noise-aware inference

Instead of ignoring the noise introduced by the privacy mechanism, we propose to perform inference over the noise in the model in Figure 1 in order to produce correct posteriors regardless of the data size. The biggest change from non-private to private Bayesian linear regression is that due to privacy constraints we can no longer condition on the covariate data $X$. The non-private posterior is $p(\boldsymbol{\theta}, \sigma^2 | X, \mathbf{y}) \propto p(\boldsymbol{\theta}, \sigma^2) \, p(\mathbf{y} | X, \boldsymbol{\theta}, \sigma^2)$ while the private posterior is $p(\boldsymbol{\theta}, \sigma^2 | \mathbf{z}) \propto \int p(X) p(\boldsymbol{\theta}, \sigma^2) \, p(\mathbf{y} | X, \boldsymbol{\theta}, \sigma^2) \, p(\mathbf{z} | X, \mathbf{y}) \, dX \, d\mathbf{y}$ (see derivations in supplementary material). The private posterior contains the term $p(X)$, which means that in order to calculate it *we need to know something about the distribution of $X$*!

Given an explicitly specified prior $p(X)$, we can perform inference over the model in Figure 1 using general-purpose MCMC algorithms. We use the No-U-Turn Sampler [Hoffman and Gelman, 2014] from the PyMC3 package [Salvatier et al., 2016], and call this method *noise-aware individual-based inference* (MCMC-Ind). This approach is simple to implement using existing tools but places a substantial burden on the modeler relative to the non-private case by requiring an explicit prior distribution $p(X)$, with poor choices potentially leading to incorrect inferences. Additionally, because MCMC-Ind instantiates latent variables for each individual, its runtime scales with population size and it may be slow for large populations.

## 3.4 Sufficient statistics-based inference

An appealing possibility is to marginalize out the variables $X$ and $\mathbf{y}$ representing individuals and instead perform inference directly over the latent sufficient statistics $\mathbf{s}$. The joint distribution is $p(\boldsymbol{\theta}, \sigma^2, \mathbf{s}, \mathbf{z}) = p(\boldsymbol{\theta}, \sigma^2) \, p(\mathbf{s} \mid \boldsymbol{\theta}, \sigma^2) \, p(\mathbf{z} \mid \mathbf{s})$. The goal is to compute a representation of $p(\boldsymbol{\theta}, \sigma^2 \mid \mathbf{z}) \propto \int_{\mathbf{s}} p(\boldsymbol{\theta}, \sigma^2, \mathbf{s}, \mathbf{z}) \, d\mathbf{s}$ by integrating over the sufficient statistics. Because this distribution cannot be written in closed form we develop a Gibbs sampler to sample from the posterior as done by Bernstein and Sheldon [2018] for unconditional exponential family models. This requires methods to sample from the conditional distributions for both the parameters $(\boldsymbol{\theta}, \sigma^2)$ and the sufficient statistics $\mathbf{s}$ given all other variables. The full conditional $p(\boldsymbol{\theta}, \sigma^2 \mid \mathbf{s})$ for the model parameters can be computed and sampled using conjugacy, exactly as in the non-private case. The full conditional for $\mathbf{s}$ factors into two terms: $p(\mathbf{s} \mid \boldsymbol{\theta}, \sigma^2, \mathbf{z}) \propto p(\mathbf{s} \mid \boldsymbol{\theta}, \sigma^2) \, p(\mathbf{z} \mid \mathbf{s})$. The first is the distribution over sufficient statistics of the regression model, for which we develop an asymptotically correct normal approximation. The second is the noise model due to the privacy mechanism, for which we use variable augmentation to ensure it is possible to sample from the full conditional distribution of $\mathbf{s}$.

### 3.4.1 Normal approximation of $\mathbf{s}$

The conditional distribution over the sufficient statistics given the model parameters is

$$p(\mathbf{s} \mid \boldsymbol{\theta}, \sigma^2) = \int_{t^{-1}(\mathbf{s})} p\left(X, \mathbf{y} \mid \boldsymbol{\theta}, \sigma^2\right) \, dX \, d\mathbf{y}, \qquad t^{-1}(\mathbf{s}) := \left\{X, \mathbf{y} : t(X, \mathbf{y}) = \mathbf{s}\right\}.$$

The integral over $t^{-1}(\mathbf{s})$, all possible populations which have sufficient statistics $\mathbf{s}$, is intractable to compute. Instead we observe that the components of $\mathbf{s} = \sum_i t(\mathbf{x}^{(i)}, y^{(i)})$ are sums over individuals. Therefore, using the central limit theorem (CLT), we approximate their distribution as $p(\mathbf{s} \mid \boldsymbol{\theta}, \sigma^2) \approx \mathcal{N}(\mathbf{s}; n\boldsymbol{\mu}_t, n\Sigma_t)$, where $\boldsymbol{\mu}_t = \mathbb{E}[t(\mathbf{x}, y)]$ and $\Sigma_t = \text{Cov}(t(\mathbf{x}, y))$ are the mean and covariance of the function $t(\mathbf{x}, y)$ on a single individual, This approximation is asymptotically correct, i.e., $\frac{1}{\sqrt{n}}(\mathbf{s} - n\boldsymbol{\mu}_t) \xrightarrow{D} \mathcal{N}(0, \Sigma_t)$ [Bickel and Doksum, 2015]. We write the conditional distribution as

$$\mathbf{s} \mid \cdot \sim \mathcal{N}(n\boldsymbol{\mu}_t, n\Sigma_t),$$

$$\boldsymbol{\mu}_t = \left[ \mathbb{E}\left[\mathrm{vec}(\mathbf{x}\mathbf{x}^T)\right], \mathbb{E}\left[\mathbf{x}y\right], \mathbb{E}\left[y^2\right] \right], \tag{2}$$

$$\Sigma_t = \begin{bmatrix} \mathrm{Cov}\left(\mathrm{vec}(\mathbf{x}\mathbf{x}^T)\right) & \mathrm{Cov}\left(\mathrm{vec}(\mathbf{x}\mathbf{x}^T), \mathbf{x}^T y\right) & \mathrm{Cov}\left(\mathrm{vec}(\mathbf{x}\mathbf{x}^T), y^2\right) \\ \mathrm{Cov}\left(\mathbf{x}y, \mathrm{vec}(\mathbf{x}\mathbf{x}^T)\right) & \mathrm{Cov}\left(\mathbf{x}y\right) & \mathrm{Cov}\left(\mathbf{x}y, y^2\right) \\ \mathrm{Cov}\left(y^2, \mathrm{vec}(\mathbf{x}\mathbf{x}^T)\right) & \mathrm{Cov}\left(y^2, \mathbf{x}y\right) & \mathrm{Var}\left(y^2\right) \end{bmatrix}. \tag{3}$$

The components of $\boldsymbol{\mu}_t$ and $\Sigma_t$ can be written in terms of the model parameters $(\boldsymbol{\theta}, \sigma^2)$ and the second and fourth non-central moments of $\mathbf{x}$ as shown below, where we have defined $\eta_{ij} := \mathbb{E}\left[x_i x_j\right]$, $\eta_{ijkl} := \mathbb{E}\left[x_i x_j x_k x_l\right]$, and $\xi_{ij,kl} := \mathrm{Cov}\left(x_i x_j, x_k x_l\right) = \eta_{ijkl} - \eta_{ij}\eta_{kl}$. Full derivations can be found in the supplementary material. We call this family of methods `Gibbs-SS`.

$$\mathbb{E}\left[x_i y\right] = \sum_j \theta_j \eta_{ij}$$

$$\mathbb{E}\left[y^2\right] = \sigma^2 + \sum_{i,j} \theta_i \theta_j \eta_{ij}$$

$$\mathrm{Cov}\left(x_i x_j, x_k y\right) = \sum_l \theta_l \xi_{ij,kl}$$

$$\mathrm{Cov}\left(x_i x_j, y^2\right) = \sum_{k,l} \theta_k \theta_l \xi_{ij,kl}$$

$$\mathrm{Cov}\left(x_i y, x_j y\right) = \sigma^2 \eta_{ij} + \sum_{k,l} \theta_k \theta_l \xi_{ij,kl}$$

$$\mathrm{Cov}\left(x_i y, y^2\right) = \sum_{j,k,l} \theta_j \theta_k \theta_l \xi_{ij,kl} + 2\sigma^2 \sum_j \theta_j \eta_{ij}$$

$$\mathrm{Var}\left(y^2\right) = 2\sigma^4 + \sum_{i,j,k,l} \theta_i \theta_j \theta_k \theta_l \xi_{ij,kl} + 4\sigma^2 \sum_{i,j} \theta_i \theta_j \eta_{ij}$$

To use this normal distribution for sampling, we need the parameters $(\boldsymbol{\theta}, \sigma^2)$ and the moments $\eta_{ij}$, $\eta_{ijkl}$, and $\xi_{ij,kl}$. The current parameter values are available within the sampler, but the modeler must provide estimates for the moments of $\mathbf{x}$, either using prior knowledge or by (privately) estimating the moments from the data. We discuss three specific possibilities in Section 3.4.4.

Once again, more modeling assumptions are needed than in the non-private case, where it is possible to condition on $\mathbf{x}$. `Gibbs-SS` requires milder assumptions (second and fourth moments), however, than `MCMC-Ind` (a full prior distribution).

### 3.4.2 Variable augmentation for $p(\mathbf{z} \mid \mathbf{s})$

The above approximation for the distribution over sufficient statistics means the full conditional distribution involves the product of a normal and a Laplace distribution,

$$p(\mathbf{s} \mid \theta, \mathbf{z}) \propto \mathcal{N}(\mathbf{s}; n\boldsymbol{\mu}_t, n\Sigma_t) \cdot \mathrm{Lap}(\mathbf{z}; \mathbf{s}, \Delta_{\mathbf{s}}/\epsilon).$$

It is unclear how to sample from this distribution directly. A similar situation arises in the Bayesian Lasso, where it is solved by variable augmentation [Park and Casella, 2008]. Bernstein and Sheldon [2018] adapted the variable augmentation scheme to private inference in exponential family models. We take the same approach here, and represent a Laplace random variable as a scale mixture of normals. Specifically, $l \sim \mathrm{Lap}(u, b)$ is identically distributed to $l \sim \mathcal{N}(u, \omega^2)$ where the variance $\omega^2 \sim \mathrm{Exp}\left(1/(2b^2)\right)$ is drawn from the exponential distribution (with density $1/(2b^2) \exp\left(-\omega^2/(2b^2)\right)$). We augment separately for each component of the vector $\mathbf{z}$ so that $\mathbf{z} \sim \mathcal{N}\left(\mathbf{s}, \mathrm{diag}(\omega^2)\right)$, where $\omega_j^2 \sim \mathrm{Exp}\left(\epsilon^2/(2\Delta_{\mathbf{s}}^2)\right)$. The augmented full conditional $p(\mathbf{s} \mid \theta, \mathbf{z}, \omega)$ is a product of two multivariate normal distributions, which is itself a multivariate normal distribution.

| **Algorithm 1** Gibbs Sampler | **Subroutine** NormProduct |
|---|---|

**Algorithm 1** Gibbs Sampler

1: Initialize $\boldsymbol{\theta}, \sigma^2, \omega^2$
2: **repeat**
3:      Calculate $\boldsymbol{\mu}_t$ and $\Sigma_t$ via Eqs. (2) and (3)
4:      $\mathbf{s} \sim \text{NormProduct}\left(n\boldsymbol{\mu}_t, n\Sigma_t, \mathbf{z}, \text{diag}(\omega^2)\right)$
5:      $\boldsymbol{\theta}, \sigma^2 \sim \text{NIG}(\boldsymbol{\theta}, \sigma^2; \boldsymbol{\mu}_n, \boldsymbol{\Lambda}_n, a_n, b_n)$ via Eqn. (1)
6:      $1/\omega_j^2 \sim \text{InverseGaussian}\left(\frac{\epsilon}{\Delta_{\mathbf{s}}|\mathbf{z}-\mathbf{s}|}, \frac{\epsilon^2}{\Delta_{\mathbf{s}}^2}\right)$ for all $j$

**Subroutine** NormProduct

1: **input:** $\boldsymbol{\mu}_1, \Sigma_1, \boldsymbol{\mu}_2, \Sigma_2$
2: $\Sigma_3 = \left(\Sigma_1^{-1} + \Sigma_2^{-1}\right)^{-1}$
3: $\boldsymbol{\mu}_3 = \Sigma_3\left(\Sigma_1^{-1}\boldsymbol{\mu}_1 + \Sigma_2^{-1}\boldsymbol{\mu}_2\right)$
4: **return:** $\mathcal{N}(\boldsymbol{\mu}_3, \Sigma_3)$

### 3.4.3 The Gibbs sampler

The full generative process is shown to the right, and the corresponding Gibbs sampler is shown in Algorithm 1. The update for $\omega^2$ follows Park and Casella [2008]; the inverse Gaussian density is $\text{InverseGaussian}(w; m, v) = \sqrt{v/(2\pi w^3)} \exp\left(-v(w-m)^2/(2m^2 w)\right)$. Note that the resulting $\mathbf{s}$ drawn from $p(\mathbf{s} \mid \boldsymbol{\mu}_t, \Sigma_t, \omega^2)$

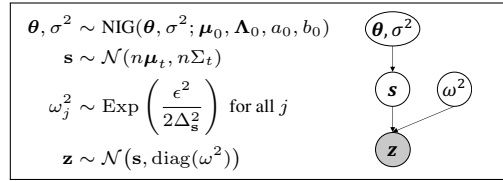

$$\boldsymbol{\theta}, \sigma^2 \sim \text{NIG}(\boldsymbol{\theta}, \sigma^2; \boldsymbol{\mu}_0, \boldsymbol{\Lambda}_0, a_0, b_0)$$
$$\mathbf{s} \sim \mathcal{N}(n\boldsymbol{\mu}_t, n\Sigma_t)$$
$$\omega_j^2 \sim \text{Exp}\left(\frac{\epsilon^2}{2\Delta_{\mathbf{s}}^2}\right) \text{ for all } j$$
$$\mathbf{z} \sim \mathcal{N}(\mathbf{s}, \text{diag}(\omega^2))$$

may require projection onto the space of valid sufficient statistics. This can be done by observing that if $A = [X, \mathbf{y}]$ then the sufficient statistics are contained in the positive-semidefinite (PSD) matrix $B = A^T A$. For a randomly drawn $\mathbf{s}$, we project if necessary so the corresponding $B$ matrix is PSD.

### 3.4.4 Distribution over $X$

As discussed above, `Gibbs-SS` requires the second and fourth population moments of $\mathbf{x}$ to calculate $\boldsymbol{\mu}_t$ and $\Sigma_t$. We propose three different options for the modeler to provide these and discuss the algorithmic considerations for each. Because we include the unit feature in $\mathbf{x}$ we can restrict our attention to the fourth moment $\mathbb{E}\left[\mathbf{x}^{\otimes 4}\right]$, which includes the second moment as a subcomponent.

**Private sample moments (`Gibbs-SS-Noisy`).** The first option is to estimate population moments privately by computing the fourth sample moments from $X$ and privately releasing them via the Laplace mechanism. The sensitivity of the estimate for $\eta_{ijkl}$ is $w_x^4$, and for $d = 2$ there are $D = 5$ unique entries, for a total sensitivity of $Dw_x^4$. This approach requires splitting the privacy budget between the release mechanisms for sufficient statistics and moments, which we do evenly. We do not perform inference over the noisy sample moments, which may introduce some miscalibration of uncertainty. Pursuing this additional layer of inference is an interesting avenue for future work.

**Moments from generic prior (`Gibbs-SS-Prior`).** A second option is to propose a prior distribution $p(\mathbf{x})$ and obtain population moments directly from the prior, either through known formulas or from Monte Carlo estimation. This approach does not access the individual data and does not consume any privacy budget, but requires proposing a prior distribution and computing the fourth moments of $\mathbf{x}$ (once) for that distribution.

**Hierarchical normal prior (`Gibbs-SS-Update`).** A final option is to perform inference over the data moments by specifying an individual-level prior $p(\mathbf{x})$ and then marginalizing away individuals, as we did for the regression model sufficient statistics. We propose a hierarchical normal prior, as shown in Figure 2a, which is more dispersed than a normal distribution and allows the modeler to propose vague priors, but still permits attainable conditional updates. The data $\mathbf{x}$ is normally distributed: $\mathbf{x} \sim \mathcal{N}(\boldsymbol{\mu}_x, \tau^2)$, with parameters drawn from the normal-inverse Wishart (NIW) conjugate prior distribution, $\boldsymbol{\mu}_x, \tau^2 \sim \text{NIW}(\boldsymbol{\mu}_0', \Lambda_0', \Psi_0', \nu_0')$. After marginalizing individuals, the latent quantities are the sufficient statistics $XX^T$ (which includes the sample mean and covariance because of the unit feature). For fixed parameters $(\boldsymbol{\mu}_x, \tau^2)$ the distribution $p(\mathbf{x})$ is multivariate normal, and we calculate its fourth moments as the fourth derivative (via automatic differentiation) of its moment generating function.

However, we introduced the new latent variables $\boldsymbol{\mu}_x$ and $\tau^2$ into the full model (see Figure 2a) and must now derive conditional updates for them within the Gibbs sampler. Naively marginalizing $X$

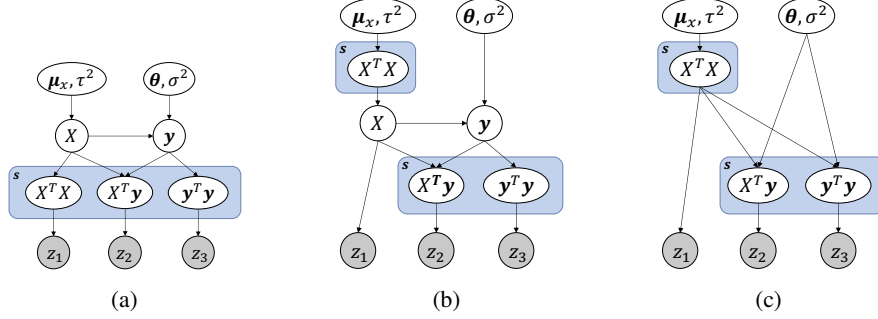

Figure 2: (a) Private Bayesian linear regression model with hierarchical normal data prior. (b) Alternative data model configuration and (c) with individual variables marginalized out.

and $\mathbf{y}$ from the full model in Figure 2a would cause both $(\boldsymbol{\mu}_x, \tau^2)$ and $(\boldsymbol{\theta}, \sigma^2)$ to be parents of $\mathbf{s}$ and thus *not* conditionally independent given $\mathbf{s}$—this would require their updates to be coupled and we could no longer use simple conjugacy formulas for each component of the model. To avoid this issue, we reformulate the joint distribution represented as in Figure 2b. The justification for this is as follows. Because $X^T X$ is a sufficient statistic for $p(X)$ under a normal model, we can encode the generative process *either* as $(\boldsymbol{\mu}_x, \tau^2) \to X \to X^T X$ *or* as $(\boldsymbol{\mu}_x, \tau^2) \to X^T X \to X$. In general, the latter formulation would require an arrow from $(\boldsymbol{\mu}_x, \tau^2)$ to $X$; this drops precisely because $X^T X$ is a sufficient statistic [Casella and Berger, 2002]. Then, upon marginalizing $X$ and $\mathbf{y}$, we obtain the model in Figure 2c. The two sets of parameters are now conditionally independent given the sufficient statistics $\mathbf{s}$, and can be updated independently as standard conjugate updates.

# 4   Experiments

We design experiments to measure the *calibration* and *utility* of the private methods. Calibration measures how close the computed posterior is to $p(\boldsymbol{\theta}, \sigma^2 | \mathbf{z})$, the correct posterior given noisy statistics. *Utility* measures how close the computed posterior is to the non-private posterior $p(\boldsymbol{\theta}, \sigma^2 | \mathbf{s})$.

## 4.1   Methods

The noise-aware individual-based method (`MCMC-Ind`) is implemented using PyMC3 [Salvatier et al., 2016]; it runs with 500 burnin iterations and collects 2000 posterior samples. The three flavors of noise-aware sufficient statistic-based methods use noisy sample moments (`Gibbs-SS-Noisy`), use moments sampled from a data prior (`Gibbs-SS-Prior`), and use an updated hierarchical normal prior (`Gibbs-SS-Update`); all three collect 20000 posterior samples after 5000 and 20000 burnin iterations for $n \in [10, 100]$ and $n = 1000$, respectively. We compare against the baseline noise-naive method (`Naive`) and the non-private posterior (`Non-Private`); both collect 2000 posterior samples.

## 4.2   Evaluation on synthetic data

**Evaluation measures.** We adapt a method of Cook et al. [2006] to measure calibration. Consider a model $p(\boldsymbol{\beta}, \mathbf{w}) = p(\boldsymbol{\beta}) p(\mathbf{w} | \boldsymbol{\beta})$. If $(\boldsymbol{\beta}', \mathbf{w}') \sim p(\boldsymbol{\beta}, \mathbf{w})$, then, for any $j$, the quantile of $\beta'_j$ in the true posterior $p(\beta_j | \mathbf{w}')$ is a uniform random variable. We can check our approximate posterior $\hat{p}$ by computing the quantile $u_j$ of $\beta'_j$ in $\hat{p}(\beta_j | \mathbf{w}')$ and testing for uniformity of $u_j$ over $M$ trials. We test for uniformity using the Kolmogorov-Smirnov (KS) goodness-of-fit test [Massey Jr., 1951]. The KS-statistic is the maximum distance between the empirical CDF of $u_j$ and the uniform CDF; lower values are better and zero corresponds to perfect uniformity, meaning $\hat{p}$ is exact.

While this test is elegant, it requires that parameters and data are drawn from the model used by the method. We use $\boldsymbol{\theta}, \sigma^2 \sim \text{NIG}\left([0, 0], \text{diag}\left(\left[\frac{.5}{20-1}, \frac{.5}{20-1}\right]\right), 20, .5\right)$. In addition, for `Gibbs-SS-Prior` and `Gibbs-SS-Update`, the test requires the covariate data be drawn from the data prior used by the methods. We specify $\boldsymbol{\mu}_x, \tau^2 \sim \text{NIW}(0, 1, 1, 50)$ and $\mathbf{x} \sim \mathcal{N}(\boldsymbol{\mu}_x, \tau^2)$. These ensure at least 95% of $\mathbf{x}$ and $y$ values are within $[-1, 1]$. We compute sensitivity assuming data

bounded in this range, but do not enforce it to avoid changing the generative process (a limitation of the evaluation method, not the inference routine). For each combination of $n$ and $\epsilon$ we run $M = 300$ trials. We qualitatively assess calibration with the empirical CDFs, which is also the *quantile-quantile* (QQ) plot between the empirical distribution of $u_j$ and the uniform distribution. A diagonal line indicates thats $u_j$ is perfectly uniform.

Between two calibrated posteriors, the tighter posterior will provide higher utility.[3] We evaluate utility as *closeness to the non-private posterior*, which we measure with *maximum mean discrepancy* (MMD), a kernel-based statistical test to determine if two sets of samples are drawn from different distributions [Gretton et al., 2012]. Given $m$ i.i.d. samples $(p, q) \sim P \times Q$, an unbiased estimate of the MMD is

$$\text{MMD}^2(P, Q) = \frac{1}{m(m-1)} \sum_{i \neq j}^{m} \left( k(p_i, p_j) + k(q_i, q_j) - k(p_i, q_j) - k(p_j, q_i) \right),$$

where $k$ is a continuous kernel function; we use a standard normal kernel. The higher the value the more likely the two samples are drawn from different distributions, therefore lower MMD between `Non-Private` and the method indicates higher utility.

We measure method runtime as the average process time over the 300 trials. Note that PyMC3 provides parallelization; we report total process time across all chains for `MCMC-Ind`.

**Results.** Calibration results are shown in Figures 3a and 3b. The QQ plot for $n = 10$ and $\epsilon = 0.1$ is shown in Figure 3c. Coverage results for 95% credible intervals are shown in Figure 3d. All four noise-aware methods are at or near the calibration-level of the non-private method, and better than `Naive`'s calibration, regardless of data size. As expected, `Gibbs-SS-Noisy` suffers slight miscalibration from not accounting for the noise injected into the privately released fourth data moment. There is slight miscalibration in certain settings and parameters for `Gibbs-SS-Prior` due to approximations in the calculation of multivariate normal distribution fourth moments from a data prior. Utility results are shown in Figure 3e; the noise-aware methods provide at least as good utility as `Naive`.

*Running time.* Figure 3f shows running time as a function of population size. We see that `MCMC-Ind` scales with increasing population size, and in fact is prohibitive to run at sizes significantly larger than $n = 100$, while all variants of `Gibbs-SS` are constant with respect to population size. It is also possible to analytically derive the asymptotic running time with respect to covariate dimension $d$. The most expensive operation used by `Gibbs-SS` will be the inversion of the covariance matrix (defined in Equation 3) in the `NormProduct` subroutine on Line 4 of Algorithm 1. This matrix has dimension $(d^2 + d + 1) \times (d^2 + d + 1)$, where $d^2 + d + 1$ are the total number of components in $t(\mathbf{x}, y) = [\mathbf{x}\mathbf{x}^T, \mathbf{x}y, y^2]$. Cubic matrix inversion would cost $O((d^2 + d + 1)^3) = O(d^6)$. Modern computing platforms can reasonably invert matrices of size 10K or more, corresponding to linear regression with $d \approx 100$ features.

## 4.3   Predictive posteriors on real data

We evaluate the predictive posteriors of the methods on a real world data set measuring the effect of drinking rate on cirrhosis rate.[4] We scale both covariate and response data to $[0, 1]$.[5] There are 46 total points, which we randomly split into 36 training examples and 10 test points for each trial. After preliminary exploration to gain domain knowledge, we set a reasonable model prior of $\boldsymbol{\theta}, \sigma^2 \sim \text{NIG}\big([1, 0], \text{diag}([.25, .25]), 20, .5\big)$. We draw samples $\boldsymbol{\theta}^{(k)}, \sigma_k^2$ from the posterior given training data, and then form the posterior predictive distribution for each test point $y_i$ from these samples.

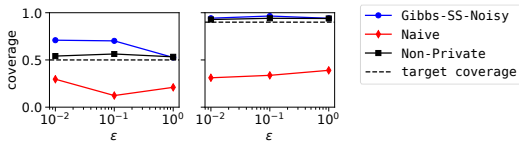

Figure 4: Coverage for predictive posterior 50% and 90% credible intervals.

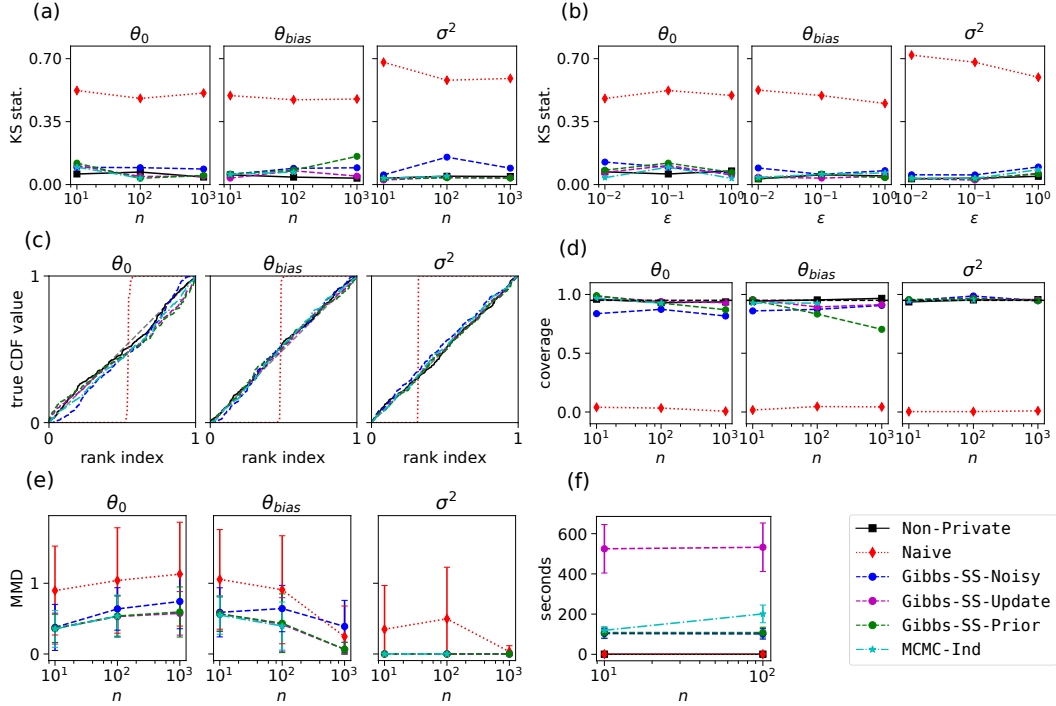

Figure 3: Synthetic data results: (a) calibration vs. $n$ for $\epsilon = 0.1$; (b) calibration vs. $\epsilon$ for $n = 10$; (c) QQ plot for $n = 10$ and $\epsilon = 0.1$; (d) 95% credible interval coverage; (e) MMD of methods to non-private posterior; (f) method runtimes for $\epsilon = 0.1$.

Figure 4 shows coverage of 50% and 90% credible intervals on 1000 test points collected over 100 random train-test splits. Non-Private achieves nearly correct coverage, with the discrepancy due to the fact that the data is not actually drawn from the prior. Gibbs-SS-Noisy achieves nearly the coverage of Non-Private, while Naive is drastically worse in this regime. We note that this experiment emphasizes the advantage of Gibbs-SS-Noisy not needing an explicitly defined data prior, as it only requires the same parameter prior that is needed in non-private analysis.

## 5 Conclusion

In this work we developed methods to perform Bayesian linear regression in a differentially private way. Our algorithms use sufficient-statistic perturbation as a release mechanism, followed by specially-designed Markov chain Monte Carlo techniques to sample from the posterior distribution given noisy sufficient statistics. Unlike in the non-private case, we cannot condition on covariates, so some assumptions about the covariate distribution are required. We proposed methods that require only moments of this distribution, and evaluated several ways to obtain the needed moments within the sampling routine.

Our algorithms are the first specifically designed for the task of Bayesian linear regression, and the first to properly account for the noise mechanism during inference. Our inferred posterior distributions are well calibrated, and are better in terms of both calibration and utility than naive SSP, which is considered a state-of-the-art baseline.

Our evaluation focused on calibration and utility of the posterior. Future work could evaluate the quality of point estimates obtained as a byproduct of our fully Bayesian algorithms. We expect such point estimates to be as good as or better than those of naive SSP, which is state-of-the-art for private linear regression [Wang, 2018]. Compared with prior work using naive SSP for linear regression, our methods are Bayesian, and perform inference over the noise mechanism. Being Bayesian is not expected to hurt point estimation. Inference over the noise mechanism is expected to not hurt, and potentially improve, point estimation.

## Footnotes

[1]This variant is called *bounded* differential privacy in that the number of individuals $n$ remains constant [Kifer and Machanavajjhala, 2011].

[2]Note that $X^T X$ is symmetric.

[3]Note that the prior itself is a calibrated distribution.

[4]http://people.sc.fsu.edu/~jburkardt/datasets/regression/x20.txt

[5]This step is not differentially private, but is standard in existing work. A reasonable assumption is that data bounds are a priori available due to domain knowledge.

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
