[Supplementary Material]



Figure 5: (a) Non-private and (b) private regression models.

# A  Appendix

## A.1  Derivation of non-private and private posteriors in Section 3.3

See corresponding models in Figure 5.

$$
\begin{aligned}
p(\boldsymbol{\theta}, \sigma^2 \mid X, \mathbf{y}) &= \frac{p(\boldsymbol{\theta}, \sigma^2, X, \mathbf{y})}{p(X, \mathbf{y})} \\
&= \frac{p(X)p(\boldsymbol{\theta}, \sigma^2)p(\mathbf{y}|X, \boldsymbol{\theta}, \sigma^2)}{p(X)p(\mathbf{y} \mid X)} \\
&= \frac{p(\boldsymbol{\theta}, \sigma^2)p(\mathbf{y}|X, \boldsymbol{\theta}, \sigma^2)}{p(\mathbf{y} \mid X)} \\
&= \frac{p(\boldsymbol{\theta}, \sigma^2)p(\mathbf{y}|X, \boldsymbol{\theta}, \sigma^2)}{\int p(\mathbf{y}, \boldsymbol{\theta}, \sigma^2 \mid X) \, d\boldsymbol{\theta}, \sigma^2} \\
&= \frac{p(\boldsymbol{\theta}, \sigma^2)p(\mathbf{y}|X, \boldsymbol{\theta}, \sigma^2)}{\int p(\boldsymbol{\theta}, \sigma^2)p(\mathbf{y} \mid X, \boldsymbol{\theta}, \sigma^2) \, d\boldsymbol{\theta}, \sigma^2}
\end{aligned}
$$

$$
\begin{aligned}
p(\boldsymbol{\theta}, \sigma^2 \mid z) &= \int p(X, \mathbf{y}, \boldsymbol{\theta}, \sigma^2, z) \, dX \, d\mathbf{y} \\
&= \int \frac{p(X, \mathbf{y}, \boldsymbol{\theta}, \sigma^2)p(z \mid X, \mathbf{y}, \boldsymbol{\theta}, \sigma^2)}{p(z)} \, dX \, d\mathbf{y} \\
&= p(z) \int p(X, \mathbf{y}, \boldsymbol{\theta}, \sigma^2)p(z \mid X, \mathbf{y}, \boldsymbol{\theta}, \sigma^2) \, dX \, d\mathbf{y}
\end{aligned}
$$

 ## A.2  Gibbs Sufficient Statistic Update

 ### A.2.1  Derivations of Equation (2): Components of $\boldsymbol{\mu}_t$

$$
\begin{aligned}
\mathbb{E}\left[x_i y\right] &= \mathbb{E}_x\left[x_i \mathbb{E}_{y|x}\left[y\right]\right] \\
&= \mathbb{E}_x\left[x_i \boldsymbol{\theta}^T \mathbf{x}\right] \\
&= \mathbb{E}_x\left[x_i \sum_j \theta_j x_j\right] \\
&= \sum_j \theta_j \mathbb{E}\left[x_i x_j\right]
\end{aligned}
$$

$$
\begin{aligned}
\mathbb{E}\left[y^2\right] &= \mathbb{E}_{\mathbf{x}}\left[\mathbb{E}_{y|\mathbf{x}}\left[y^2\right]\right] \\
&= \mathbb{E}_{\mathbf{x}}\left[\sigma^2 + \left(\boldsymbol{\theta}^T \mathbf{x}\right)^2\right] \\
&= \sigma^2 + \mathbb{E}\left[\left(\sum_i \theta_i x_i\right)^2\right] \\
&= \sigma^2 + \mathbb{E}\left[\sum_{i,j} \theta_i \theta_j x_i x_j\right] \\
&= \sigma^2 + \sum_{i,j} \theta_i \theta_j \mathbb{E}\left[x_i x_j\right]
\end{aligned}
$$

 ### A.2.2  Derivations of Equation (3): Components of $\Sigma_t$

$$
\begin{aligned}
\operatorname{Cov}\left(x_i x_j, x_k y\right) &= \mathbb{E}\left[x_i x_j x_k y\right] - \mathbb{E}\left[x_i x_j\right]\mathbb{E}\left[x_k y\right] \\
&= \mathbb{E}_x\left[x_i x_j x_k \mathbb{E}_{y|x}\left[y\right]\right] - \mathbb{E}\left[x_i x_j\right]\mathbb{E}\left[x_k y\right] \\
&= \mathbb{E}_x\left[x_i x_j x_k \sum_l \theta_l x_l\right] - \mathbb{E}\left[x_i x_j\right]\sum_l \theta_l \mathbb{E}\left[x_k x_l\right] \\
&= \sum_l \theta_l \mathbb{E}\left[x_i x_j x_k x_l\right] - \sum_l \theta_l \mathbb{E}\left[x_i x_j\right]\mathbb{E}\left[x_k x_l\right] \\
&= \sum_l \theta_l \operatorname{Cov}\left(x_i x_j, x_k x_l\right)
\end{aligned}
$$

$$
\begin{aligned}
\operatorname{Cov}\left(x_i x_j, y^2\right) &= \mathbb{E}\left[x_i x_j y^2\right] - \mathbb{E}\left[x_i x_j\right]\mathbb{E}\left[y^2\right] \\
&= \mathbb{E}_x\left[x_i x_j \mathbb{E}_{y|x}\left[y^2\right]\right] - \mathbb{E}\left[x_i x_j\right]\mathbb{E}\left[y^2\right] \\
&= \mathbb{E}_x\left[x_i x_j \left(\sigma^2 + \sum_{k,l}\theta_k \theta_l x_k x_l\right)\right] - \mathbb{E}\left[x_i x_j\right]\left(\sigma^2 + \sum_{k,l}\theta_k \theta_l \mathbb{E}\left[x_k x_l\right]\right) \\
&= \sigma^2 \mathbb{E}\left[x_i x_j\right] + \sum_{k,l}\theta_k \theta_l \mathbb{E}\left[x_i x_j x_k x_l\right] - \sigma^2 \mathbb{E}\left[x_i x_j\right] - \sum_{k,l}\theta_k \theta_l \mathbb{E}\left[x_i x_j\right]\mathbb{E}\left[x_k x_l\right] \\
&= \sum_{k,l}\theta_k \theta_l \operatorname{Cov}\left(x_i x_j, x_k x_l\right)
\end{aligned}
$$

$$\mathrm{Cov}\left(x_i y, x_j y\right) = \mathbb{E}\left[x_i x_j y^2\right] - \mathbb{E}\left[x_i y\right]\mathbb{E}\left[x_j y\right]$$

$$= \mathbb{E}_x\left[x_i x_j \mathbb{E}_{y|x}\left[y^2\right]\right] - \left(\sum_k \theta_k \mathbb{E}\left[x_i x_k\right]\right)\left(\sum_l \theta_l \mathbb{E}\left[x_j x_l\right]\right)$$

$$= \mathbb{E}\left[x_i x_j\left(\sigma^2 + \sum_{k,l}\theta_k\theta_l x_k x_l\right)\right] - \sum_{k,l}\theta_k\theta_l \mathbb{E}\left[x_i x_k\right]\mathbb{E}\left[x_j x_l\right]$$

$$= \sigma^2 \mathbb{E}\left[x_i x_j\right] + \sum_{k,l}\theta_k\theta_l\left(\mathbb{E}\left[x_i x_j x_k x_l\right] - \mathbb{E}\left[x_i x_k\right]\mathbb{E}\left[x_j x_l\right]\right)$$

$$= \sigma^2 \mathbb{E}\left[x_i x_j\right] + \sum_{k,l}\theta_k\theta_l\,\mathrm{Cov}\left(x_i x_k, x_j x_l\right)$$

$$\mathrm{Cov}\left(x_i y, y^2\right) = \mathbb{E}\left[x_i y^3\right] - \mathbb{E}\left[x_i y\right]\mathbb{E}\left[y^2\right]$$

$$= \mathbb{E}_x\left[x_i \mathbb{E}_{y|x}\left[y^3\right]\right] - \mathbb{E}\left[x_i y\right]\mathbb{E}\left[y^2\right]$$

$$= \mathbb{E}_x\left[x_i\left(\sum_{j,k,l}\theta_j\theta_k\theta_l x_j x_k x_l + 3\sigma^2\sum_j \theta_j x_j\right)\right]$$

$$- \sum_j \theta_j \mathbb{E}\left[x_i x_j\right]\left(\sigma^2 + \sum_{k,l}\theta_k\theta_l x_k x_l\right)$$

$$= \sum_{j,k,l}\theta_j\theta_k\theta_l \mathbb{E}\left[x_i x_j x_k x_l\right] + 3\sigma^2\sum_j \theta_j \mathbb{E}\left[x_i x_j\right]$$

$$- \sigma^2\sum_j \theta_j \mathbb{E}\left[x_i x_j\right] + \sum_{j,k,l}\theta_j\theta_k\theta_l \mathbb{E}\left[x_i x_j\right]\mathbb{E}\left[x_k x_l\right]$$

$$= \sum_{j,k,l}\theta_j\theta_k\theta_l\,\mathrm{Cov}\left(x_i x_j, x_k x_l\right) + 2\sigma^2\sum_j \theta_j \mathbb{E}\left[x_i x_j\right]$$

$$\mathrm{Var}\left(y^2\right) = \mathbb{E}\left[y^4\right] - \mathbb{E}\left[y^2\right]^2$$

$$= 3\sigma^4 + \sum_{j,k,l,m}\theta_j\theta_k\theta_l\theta_m \mathbb{E}\left[x_j x_k x_l x_m\right] + 6\sigma^2\sum_{j,k}\theta_j\theta_k \mathbb{E}\left[x_j x_k\right] - \left(\sigma^2 + \sum_{j,k}\theta_j\theta_k \mathbb{E}\left[x_j x_k\right]\right)^2$$

$$= 3\sigma^4 + \sum_{j,k,l,m}\theta_j\theta_k\theta_l\theta_m \mathbb{E}\left[x_j x_k x_l x_m\right] + 6\sigma^2\sum_{j,k}\theta_j\theta_k \mathbb{E}\left[x_j x_k\right]$$

$$- \sigma^4 - 2\sigma^2\sum_{j,k}\theta_j\theta_k \mathbb{E}\left[x_j x_k\right] - \sum_{j,k,l,m}\theta_j\theta_k\theta_l\theta_m \mathbb{E}\left[x_j x_k\right]\mathbb{E}\left[x_l x_m\right]$$

$$= 2\sigma^4 + \sum_{j,k,l,m}\theta_j\theta_k\theta_l\theta_m\left(\mathbb{E}\left[x_j x_k x_l x_m\right] - \mathbb{E}\left[x_j x_k\right]\mathbb{E}\left[x_l x_m\right]\right) + 4\sigma^2\sum_{j,k}\theta_j\theta_k \mathbb{E}\left[x_j x_k\right]$$

$$= 2\sigma^4 + \sum_{j,k,l,m}\theta_j\theta_k\theta_l\theta_m\,\mathrm{Cov}\left(x_j x_k, x_l x_m\right) + 4\sigma^2\sum_{j,k}\theta_j\theta_k \mathbb{E}\left[x_j x_k\right]$$