[Reviews · NeurIPS 2019]

Reviewer 1



This paper is methodological (and experimental) in nature, providing a suite of approaches to differentially-private Bayesian linear regression. The key significance is to revisit DP linear regression in the Bayesian setting, where it is natural to consider 1) how privacy-preserving noise affects posterior estimates; 2) leverage Bayesian inference through directly modelling the noise process, to improve utility (broadly construed including in terms of calibration). The paper does a quality job of exploring how such modelling and inference could be performed based on sufficient statistic perturbation. The paper has high clarity, further adding to the potential practical impact. The main technical ideas are largely inspired by prior work such as Bernstein and Sheldon (2018)'s work on exponential families. Although it is important to note that the extension does require careful technical work. There are some improvements listed below that could have led to a more comprehensive treatment. Some minor comments: line 20 and in related work otherwise, the discussion around posterior sampling being deficient in forming private posteriors seems at odds with the fundamental approach taken of MCMC - also sampling from the posterior. I view the key distinction here as instead modelling the DP noise process akin to the vision set out by Williams and McSherry [34] and applied here with care. Def 2 line 68 the max might not exist, instead use supremum (and Delta could be an upper bound). It is appropriate to cite McSherry & Mironov's KDD'09 paper as it is early work perturbing sufficient statistics for DP. The feature normalisation in line 304 - I don't think that's DP, is it? UPDATE POST REBUTTAL I thank the authors for their thoughtful rebuttal. Their response has cleared up a number of questions around VI, feature selection (not ideal but fair point regarding convention), and more. I like the work and believe it will have practical impact. I'd second another reviewer (also mentioned in my initial review I believe) that it would be nice to see some point estimate baseline comparisons made, even though the authors could lift their rebuttal comments directly into the paper instead, I view the point as more about being comprehensive without taking much space/time to achieve - it may help the paper have further reach, and does reflect on interesting questions that I'm sure other readers may have too.

Reviewer 2



The work is a solid contribution towards refining privacy-preserving bayesian linear regression, in which the bayesian interpretation is handled correctly even after injecting privatization noise. Several methods of implementing these models are explored, and the theory behind these changes are evidenced by empirical tests. Originality: the MCMC based method appears to mainly be a direct use of a mostly off-the-shelf idea, but the derivations of the Gibbs updates for the sufficient statistics based model are novel. Related work and appropriate comparisons are cited. Quality: The work is a solid complete contribution, with propositions backed up by empirical tests. Clarity: the work is clearly written and structured. Significance: the work greatly improves over the naive baseline, and the sufficient statistics method is shown to also be nicer than the MCMC method while achieving comparable results.

Reviewer 3



Originality: The main novelty vs Bernstein and Sheldon (2018) [5] is handling the fact that regression models condition on the data, which is private, when accounting for the noise of the mechanism. This is a valuable advance, though somewhat incremental. Quality: The approach of accounting for mechanism noise in posterior uncertainty is extremely elegant (though the basic idea of that follows from [5]). The proposed MCMC approaches are sensible, and the hierarchical normal prior formulation with conditionally conjugate updates is very clever. Overall, I like the proposed ideas. The experiments are the main weakness of the current manuscript. The real and synthetic data both have only 2 dimensions, and the real datasest only has 46 data points. While this data regime does have some real-world significance, it is not exactly modern. I would like to have seen some larger-scale results. The one posterior sample (OPS) method should also be used as a baseline (although I expect that the proposed methods would beat it, especially in this regime, due to its poor data efficiency). There are probably several strong point-estimate private linear regression models in the literature which should ideally be compared to as well. Clarity: The paper is well written and easy to read. The only issue is that there is no conclusion section to wrap up, instead the paper simply stops (presumably due to running out of space). Significance: The paper addresses an important problem (differentially private Bayesian linear regression, accounting for mechanism noise) and proposes elegant solutions. Its significance would be increased if the experiments could demonstrate the methods efficacy in more realistic, higher-dimensional problems.

[Author Response · NeurIPS 2019]

Thank you to the reviewers for the constructive and positive comments. Many comments had to do with positioning
relative to existing work, which we will clarify in a revision. The comments can be broken into two groups: point
estimation methods and Bayesian methods.

**Point Estimation**

We chose to investigate the ability of methods to produce calibrated posteriors, but agree that evaluation of point
estimates is an interesting and important question. Our initial hypothesis was that noise-aware methods will in general
perform similarly to their noise-naive counterparts for point estimation. We observed this in preliminary experiments
for simple estimation problems (not regression). However, in ongoing applications of our method from this paper, we
observe that it produces better point estimates than noise-naive SSP. Furthermore, noise-naive SSP is a competitive and
often state-of-the-art baseline for point estimation (Foulds 2016, Wang 2018). We therefore plan on exploring this more
deeply in the future.

In general, a Bayesian method is not expected to produce better point estimates than a non-Bayesian counterpart unless
the prior is informative.

We also conjecture that noise-aware inference will almost never hurt when it comes to point estimation, when compared
to a similar noise-naive algorithm. We also conjecture that it will not help *in general*, which is confirmed by experiments
in simple estimation problems. However, there are two possible mechanisms by which noise-aware inference *can*
improve point estimation. One is by automatically respecting any constraints on parameters, if present. Another is
by avoiding certain pathologies of noise-naive methods. For example, for linear regression, noisy sufficient statistics
introduce bias into the traditional least-squares estimator, which is unbiased in the absence of noise introduced for
privacy. Our noise-naive Bayesian method may avoid this pathology.

**Other Bayesian Methods**

As for other Bayesian methods, all are noise-naive. We compared to naive SSP, which we consider to be the most
competitive baseline in this setting. The posterior sampling MCMC method due to Wang (ICML 2015) allows public
release of many posterior samples (unlike OPS), but still suffers from per-sample privacy loss due to the noise injected
in each iteration. Private variational inference (VI) is most relevant for problems where the original posterior inference
problem requires VI, i.e., when naive SSP is not an option. Bayesian linear regression permits conjugate updates and
therefore we can use naive SSP. In fact, VIPS due to Park et al. uses naive SSP on a per iteration basis within the
variational inference framework. It is clearly better to privatize the full sufficient statistics only once as in naive SSP.
We will clarify these points in the related work section.

**Runtime and Data Size**

There are two parameters that may affect runtime: number of individuals and number of dimensions. The plot in Figure
3f addresses the former. Its purpose is to emphasize that the runtime of `MCMC-Ind` scales with the number of individuals,
but the runtime of `Gibbs-SS` methods do not, due to their usage of fixed-dimension sufficient statistics. We will clarify
that point in the text. The main purpose of the plot was not to compare against `Naive` or to make a statement about
runtime for specific practical settings, though that is a question of interest.

While we have not run in-depth experiments on running time relative to covariate dimensionality, we can understand
its effect analytically. Let $d$ be the covariate dimension. The most expensive operation used by `Gibbs-SS` will be the
inversion of the covariance matrix (defined in Equation 3) in the NormProduct subroutine on Line 4 of Algorithm 1.
This matrix has dimension $(d^2 + d + 1) \times (d^2 + d + 1)$, where $d^2 + d + 1$ are the total number of components in the
feature function vector $t(\mathbf{x}, y) = [\mathbf{x}\mathbf{x}^T, \mathbf{x}y, y^2]$. The cubic matrix inversion would then be $O((d^2 + d + 1)^3) = O(d^6)$.
Modern computing platforms can reasonably invert matrices of size 10K or more, corresponding to linear regression
with $d \approx 100$ features.

**Misc.**

We agree with R1's suggestion to cite McSherry & Mironov's KDD'09 paper as an early work on SSP.

R1 correctly notes that the normalization of the $X$ and $y$ data in the Section 4.3 case study is in fact not a private
operation and the normalization constant would have to be publicly released in order to make predictions on new
instances. We found this step to be a preprocessing assumption prevalent in many existing works, e.g. the Jalko et
al. paper previously mentioned by R3, and one could assume these bounds come from domain knowledge instead of
sensitive private data.

[Meta-Review · NeurIPS 2019]

All reviewers consider the paper to make a solid contribution and recommend acceptance.